# Microbiota in the ptarmigan intestine—An Inuit delicacy and its potential in popular cuisine

Mads Bjørn Bjørnsen [1,2]*, Nabila Rodríguez Valerón [3], Diego Prado Vásquez[4], Esther Merino Velasco [3,4], Anders Johannes Hansen[2,5], Aviaja Lyberth Hauptmann [1,5]*

1 SILA Department, Institute of Health and Nature, Ilisimatusarfik—University of Greenland, Nuuk, Greenland, 2 Section for Geogenetics, Globe Institute, University of Copenhagen, Copenhagen, Denmark, 3 Basque Culinary Center, Facultad de Ciencias Gastronomicas, Mondragon Unibertsitatea Donostia, San Sebastian, Spain, 4 TABA Project, Research & Development Studio, Laguna de Duero, Spain, 5 Center for Evolutionary Hologenomics, Globe Institute, University of Copenhagen, Copenhagen, Denmark

* mads.bjornsen@sund.ku.dk (MBB); alha@uni.gl (ALH)

**Data Availability Statement:** The data is available at the NCBI SRA repository with accession number PRJNA1162031.

## Abstract

The consumption of prey intestines and their content, known as gastrophagy, is well-documented among Arctic Indigenous peoples, particularly Inuit. In Greenland, Inuit consume intestines from various animals, including the ptarmigan, a small herbivorous grouse bird. While gastrophagy provides the potential to transfer a large number of intestinal microorganisms from prey to predator, including to the human gut, its microbial implications remain to be investigated. This study addresses this gap by investigating the microbial composition of the Greenlandic rock ptarmigan's gastrointestinal tract by analyzing the crop, stomach, and intestines while also comparing it with the microbiota found in garum, a fermented sauce made from ptarmigan meat and intestines. Through 16S rRNA gene sequencing, we assessed whether garum made from ptarmigan intestines provides access to microbial diversity otherwise only accessible through gastrophagy. Our findings reveal that garum made from ptarmigan intestines displayed distinct flavors and microbial composition similar to that found in the ptarmigan gut and intestines, highlighting the potential role of fermented products in mediating food microbial diversity associated with Indigenous food practices. Furthermore, our study underscores the broader importance of understanding microbial diversity in different food systems, particularly in the context of shifting dietary patterns and concerns about diminishing food microbial diversity. By elucidating the microbial richness gained through gastrophagy this research contributes to a deeper understanding of traditional and Indigenous foodways and their implications for human gut health.

## Introduction

The study of the microbiology of foods has conventionally dealt primarily with pathogenic microorganisms posing threats to food safety and quality. However, with the rapid expansion of our understanding of the human gut microbiome, there is an increasing focus on the

**Funding:** The project was funded by the following institutions: the Aage V. Jensen Foundation and the Danish National Research Foundation. ALH received: Aage V. Jensen Foundation. Granted 2022 https://avjcf.org/ AJH received Danish National Research Foundation. DNRF CEH grant DNRF143 https://dg.dk/en/ The funders had no role in study design, data collection and analysis, decision to publish, or preparation of the manuscript.

**Competing interests:** The authors have declared that no competing interests exist.

importance of food microbial diversity for human health [1]. Exposure to microbial diversity, including non-pathogenic bacteria in foods, has important implications for human gut microbiome diversity and proper functioning [2]. Fermented foods can play a crucial role in promoting gut health by introducing beneficial microbes into the digestive system [3, 4]. Despite this, except in the case of fermented foods, food microbiology studies are often uncoupled to considerations of food as a source of microbial diversity for the human gut (as exemplified by recent thematic issues on the topic of food microbial diversity [5]) with a few notable exceptions [6, 7]. Thus, there is a need to explore the microbial diversity and composition of various foods beyond the lens of food safety alone, as it has the potential to impact human health positively. As a result, we currently have a limited understanding of the gut microbial implications of other microbially rich foods. An example of such food is gastrophagy, which is the eating of intestines and intestinal content, a food practice that can potentially transfer high numbers of intestinal microbes between trophic levels in the food web. Gastrophagy is common in non-industrialized food cultures [8]. It is well-documented among the Arctic Indigenous peoples [9–12], particularly among Inuit [13]. So far, research on gastrophagy has focused on taste, culture, and nutrition [13] but not the apparent potential for food microbial diversity. Previous studies from our lab have shown that the Indigenous foods of Inuit in Greenland, made according to tradition, harbor a more diverse food microbiota than their industrial counterparts [14]. We have argued that eating intestines as part of the Inuit diet has nutritional and microbial implications for the human gut microbiota [15]. In this study, we take an additional step towards understanding the microbiological potential of Inuit microbe-rich foods by assessing the microbiota of *orunit*—the intestines and intestinal content of the ptarmigan, which has traditionally been eaten raw [16, 17]. The rock ptarmigan (*Lagopus muta*) is a small grouse species in the Arctic and subarctic areas. Traditionally, the ptarmigan was mainly treasured for its intestines, eaten raw rather than its meat [16]. Today, cooked meat is an integrated part of local diets in Greenland, while eating its intestines is rare. We included an assessment of the microbiological potential of using ptarmigan intestines in a popular gastronomic context, namely garum. Garum is a traditional fermented fish sauce from ancient Roman and Greek cuisine. It is originally made by fermenting typically small oily fishlike anchovies and is known for its rich flavor [18]. In our case, the garum was made of the ptarmigans' meat and intestines.

Previous research has focused on the microbial composition of cecal samples from wild rock ptarmigans in Arctic Norway and Svalbard [19] and compared wild Japanese rock ptarmigans and captive Svalbard rock ptarmigans [20]. *Firmicutes*, *Actinobacteria*, *Bacteroidetes*, and *Synergistetes* were shown to be the dominant phyla in the wild rock ptarmigans. In contrast, a more homogenized composition dominated by only the phyla *Firmicutes* and some *Bacteroidota* was found in the captive rock ptarmigans, which supports our earlier research showing that a richer source of microbial diversity is found in non-industrialized foods [15]. No research that we know of has delved into the composition of the Greenlandic rock ptarmigan microbiota, and no former studies have approached it from the perspective of gastrophagy and the potential microbial importance of Indigenous peoples' food cultures.

By utilizing 16S rRNA gene amplicon sequencing, we analyzed the microbial composition along the gastrointestinal tract of Greenlandic rock ptarmigan as well as the microbial composition of garum made from the ptarmigan meat and intestines. We investigated whether the microbial composition changed through the intestinal tract and whether the microbiota found in the ptarmigan's intestines was integrated into the garum microbiota. This study aimed to describe the microbial composition of Greenlandic rock ptarmigans and the potential microbial diversity available through gastrophagy. Additionally, we aimed to determine if the microbial diversity present in the intestines could be transferred to popular cuisines. By including an

assessment of the microbial potential of ptarmigan intestines in the context of popular gastronomy, namely Koji garum, we wished to highlight the relevance of ptarmigan gastrophagy not only as part of the past but also in the present and the future.

## Materials and methods

### Ethics statement

The animals included in this study–Greenlandic rock ptarmigans (*Lagopus muta*)–are not endangered or protected. The samples for this project were collected with approval from the Government of Greenland (NO. G23-018) during the regulated hunting season in the fall/winter of 2021. Greenland does not have an Institutional Animal Care and Use Committee, and no other approval beyond the sample permission listed above is required in Greenland.

### Sample collection

Samples were obtained from 11 rock ptarmigans from the Nuuk fjord system in West Greenland. Bacterial swabs were collected from the crop, stomach (gizzard), and intestine (cecum) in triplicates using sterile swabs (HAGMED®, Rawa Mazowiecka, Poland) in the autumn of 2021 and stored at -80˚C. Ptarmigan meat and intestines were used to make garum. For the garum, pearled barley (Aurion, Denmark) was soaked for 12 hours at 4˚C and then washed with water. The barley was steamed at 100˚C for 90 minutes and then cooled down until 30˚C and inoculated with *Aspergillus oryzae* (Barley white koji, Higuchi Matsunosuke Shoten Co., Ltd) for 36 hours at 31˚C and 80% humidity in an incubator (IF110plus, Memmert, GmbH + Co. KG, Schwabach, Germany).

To prepare the garum, the 11 ptarmigan birds were brought frozen to the kitchen, whereafter they were defrosted and cleaned from feathers, separating meat from intestines. Subsequently, barley koji was then mixed with either ptarmigan intestines (two samples) and meat (two samples), separately (four samples), in a ratio of 1:2 w/w (Table 1). Finally, 6% salt was added, mixed, and stored in a sealed glass jar at 28˚C for one month. Samples from the four different garums were also taken in triplicates. In total, 111 samples were taken: crop ($n = 33$), stomach ($n = 33$), intestines ($n = 33$), and garum ($n = 12$).

### Taste evaluation

The taste of garum was evaluated by 12 trained chefs and researchers from Basque Culinary Center (San Sebastian, Spain) familiar with fermented meat and fish sauce by using check-all-that-apply (CATA) with an adapted description of 33 taste attributes for fermented sauce and garum [21] S1 Table. Three ml of each garum was served in a small transparent plastic cup at room temperature 20˚C.

### DNA extraction, amplification, and sequencing

We extracted DNA from a total of 118 samples, including 111 ptarmigan samples, one negative control from an unused Type AD sterile swab (HAGMED®, Rawa Mazowiecka, Poland), and

**Table 1. The garum content is produced in duplicates from the ptarmigan intestines and meat, respectively.**

| Garum Intestines 1 | Garum Intestines 2 | Garum Meat 1 | Garum Meat 2 |
|---|---|---|---|
| 310g Barley Koji | 311g Barley koji | 800g Barley koji | 755g Barley koji |
| 621g Intestines | 621g Intestines | 1599g Meat | 1509g Meat |
| 40g Salt | 60g Salt | 154g Salt | 144g Salt |

six DNA extraction kit controls using the DNeasy PowerSoil Pro Kit (Qiagen®, Hilden, Germany) and following the manufacturer's instructions with a few modifications as described below. The swab tip was cut off with a sterile scissor and placed into the PowerBead Pro tube, where 800 µl of solution C1 was added and vortexed briefly. Hereafter, the samples were incubated at 65°C for 10 minutes. The DNA concentration was assessed by a Qubit fluorometer (Invitrogen) and stored at -20°C until further processing. A clean-up process with DNeasy PowerClean Pro Cleanup kit (Qiagen®, Hilden, Germany) and OneStep PCR Inhibitor Removal Kit (Zymo Research®, CA, U.S.A.) was performed to remove the remaining PCR inhibitors using the manufacturer's instructions.

qPCR visualization of selected samples from each sample type showed that combining both clean-up processes gave the best amplification results. The V3 and V4 regions of the 16S rRNA gene were amplified through PCR from the extracted DNA using the primers 341F: ACTCCTACGGGAGGCAGCAG [22] and 806R: GGACTACHVGGGTWTCTAAT [23]. Furthermore, two PCR setups were used as some samples did not amplify, and others were overamplified. The following PCR setup was used for the garum samples: 95°C for 10 min, followed by 32 cycles of 95°C for 15 sec, 56°C for 20 sec, and 72°C for 40 sec. This was followed by a single cycle of 72°C for 10 min, whereafter it was stored at -20°C until further processed. Samples from the crop, stomach, and intestines had the following PCR setup: 95°C for 10 min, followed by 32 cycles of 95°C for 15 sec, 56°C for 20 sec, and 72°C for 40 sec. Followed by a single cycle of 72°C for 10 min and then stored at -20°C until further processed. During this process, we added six PCR controls, leaving us with a total of 124 samples for the downstream library build. PCR products were resolved with gel electrophoresis and stored at -20°C. The PCR product was pooled to make amplicon libraries with Illumina TrueSeq DNA PCR-free library, with each PCR product equally represented by assessing the gel electrophoresis results and dividing the samples into groups depending on the results. The library was built on Ilumina TrueSeq DNA PCR-free Library following the manufacturer's instructions, and the final libraries were sent for Illumina MiSeq v3 2 x 300 bp paired-end sequencing. Negative control, DNA extraction kit controls, and PCR product controls were included throughout all stages of the laboratory process. The samples were sequenced a second time to ensure enough data for further processing.

## Bioinformatics and statistical analysis

**Pre-processing and data analysis.** For 16S rRNA gene amplicon sequencing data, raw reads were extracted from the Illumina Miseq® System in fastq format. The paired-end reads were first demultiplexed and trimmed for primers using cutadapt v. 4.2 [24]. Reads with an expected error rate -e 0.13, containing one or more Ns or a length below 170 bp, were discarded. Furthermore, each sample was demultiplexed into sense and anti-sense reads for subsequent processing, as described by Frøslev and colleagues (2017). Through the DADA2 framework [25], the sense and anti-sense reads were processed independently until the identification of chimeric sequences [26]. The algorithm was used to filter and denoise; chimeras were removed before generating the sense, and the anti-sense ASV table contained ASV read counts for each merged sample. Data from the two individual sequence runs were merged, combining the reads from both runs into one ASV table [27]. Taxonomy assignments were performed against the SILVA 16S taxonomy database v.138.1 [28]. R package Decontam v. 1.20.0 [29] was used to detect ASVs present in controls and determine the probability for those to be contaminants, where 4 ASVs were identified as such, whereafter controls and blanks were filtered out of the data. Furthermore, 190 ASVs not annotated to at least bacterial Phylum were removed from downstream analyses, and singletons and doubletons were removed from

the dataset. Seven samples with <5.000 reads, comprising 3 crops and 4 stomach samples, were removed from the dataset, leaving 104 samples for further analysis [30, 31].

**Statistics and visualization.** Statistical analysis and visualization of graphs were conducted in the "R" software RStudio v. *2023.06.0* [32]. Rarefaction curves were plotted using vegan package v. 2.6.4 [33, 34]. Student t-test was used to test for bacterial differences at the phylum and genus level. ggplot2 package v. 3.4.4 [35] was used to visualize barplots of the microbial communities found in the sample types. Alpha diversity for four diversity indices was calculated: Richness, Shannon, Simpson, and Faith phylogenetic diversity, for which a phylogenetic tree was built using package ape v. 5.7.1.4 [36]. The diversity was tested for normal distribution using the Shapiro-Wilks test. Whereafter, the non-parametric Kruskal-Wallis test was used to compare the diversity indices across the sample types for significant differences (p <0.05), with the p-value adjusted for False Discovery Rate method [37]. The R package Phyloseq v. 1.44.0 [38, 39] was used to calculate the Richness, Shannon, and Simpson indices, and the R package picante v. 1.8.2 [40] was used to calculate the phylogenetic diversity indices. The sample's core microbiome was identified using the R package Microbiome v. 1.22.0 [41]

The unweighted Unifrac dissimilarity distance matrix was used for beta diversity analysis and visualized using a Non-Metric Multi-dimensional scaling (NMDS) plot was performed on the distance matrix to visualize the dissimilarity between the microbial communities and colored by metadata variable using the phyloseq v. 1.44.0 [38, 39] and vegan packages. The R package vegan was used to compare bacterial community composition between the groups using a Permutational Multivariate Analysis of Variance (PERMANOVA) with 999 permutations using Adonis2 [33] on the matrix to determine if dissimilarity was significant between sample types pairwiseAdonis package v. 0.4.1 [42]. Statistically significant results were considered at p-value <0.05, with Bonferroni used for p-value adjustment for multiple comparisons.

Fisher's exact test was used when the Cochran Q test did not apply to the CATA data to determine whether there were significant differences between the tasting attributes associated with garum intestines and garum meat, statistically significant results were considered at p-value <0.05.

## Results

We sequenced the V3-V4 region of the 16S rRNA with Illumina Miseq v3, which yielded 32,682,297 forward and 32,682,297 reverse reads from two runs. After filtering and quality process, we ended up with 104 samples (Crop = 30), (Intestines = 33), (Stomach = 29), and (Garum = 12) retaining a total of 8.466,809 combined sequences with an average number of sequences per sample of 81,411 clustered into 3090 ASVs, which represented 2 Kingdoms, 23 phyla, 33 classes, 81 orders, 136 families, and 251 genera. ASVs accumulation curves reached asymptote, confirming sufficient sequencing depth to cover the microbial diversity in all samples (S1 Fig).

### Composition of ptarmigan intestinal microbiota

The ptarmigan gut, including crop, stomach, and intestines, harbored a microbial community of 2979 ASVs. The crop contained 809 ASVs and was dominated mainly by *Firmicutes* (68.24% ±31.2) and *Proteobacteria* (22.32% ±25.03) at the phylum level (Fig 1A). The stomach contained 1782 ASVs and was dominated by *Firmicutes* (35.75% ±10.73), *Actinobacteriota* (34.13% ±14.99), and *Proteobacteria* (10.96% ±16.55). Finally, the intestines contained 1534 ASVs and were dominated by *Campylobacterota* (24.66% ±33.39), *Firmicutes* (24.45% ±19.38), *Actinobacteriota* (22.25% ±19.53), and *Proteobacteria* (12.68% ±22.58). At the genus level, the

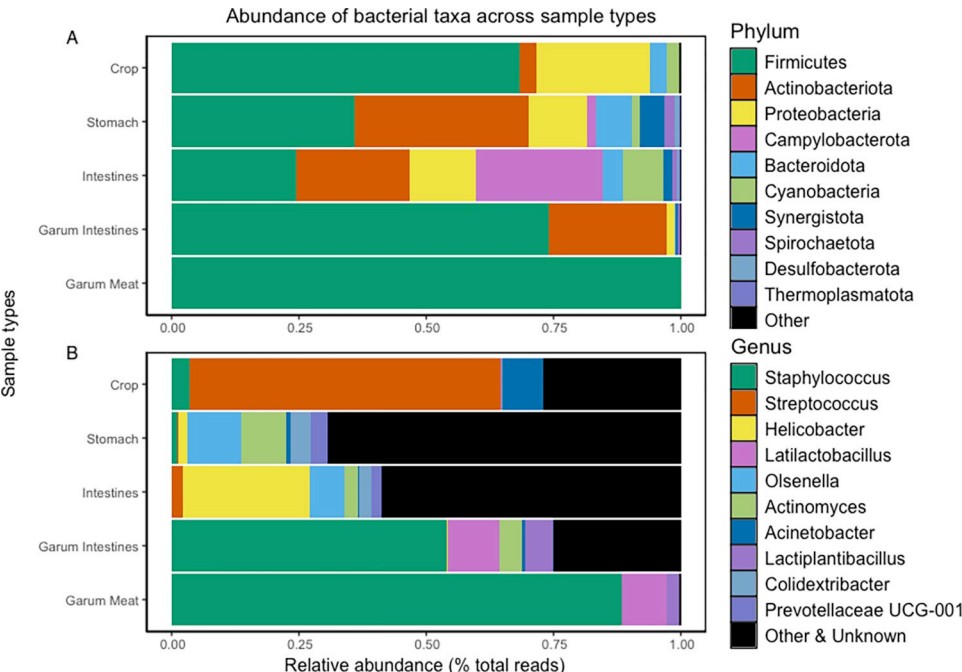

**Fig 1. Taxonomic composition across sample types at phylum and genus level.** Microbial composition comparison of the 10 most abundant taxa between sample types. Each horizontal column indicates the average microbial relative abundance within each sample type (Crop, n = 30; Stomach, n = 29; Intestines, n = 33; Garum Intestines, n = 6; Garum Meat, n = 6) at (A) phylum level (B) genus level.

crop contained mainly *Streptococcus* (61.2% ±36.57), *Acinetobacter* (8.16% ±16.79), and unknown bacteria (4.01% ±7.46) (Fig 1B). The stomach had unknown bacteria (42.46% ± 12.69), *Olsenella* (10.5% ± 9.5), and *Actinomyces* (8.99% ± 8.40). Finally, the intestines were dominated by unknown bacteria (44.48% ±24.11), *Helicobacter* (24.87% ±33.65), and *Olsenella* (6.68% ±7.33). We also analyzed the garum samples consisting of 285 ASVs, mainly comprised of two phyla, *Firmicutes* (86.77% ±16.91) and *Actinobacteriota* (11.78% ±15.08). When we divided the garum samples into the two sub-types, from intestines and meat, we saw that the garum made from intestines had 250 ASVs and, at the phylum level, mainly *Firmicutes* (73.55% ±14.48) and *Actinobacteriota* (23.55% ±12.94). In contrast, the garum made from meat consisted of only 66 ASVs. It was mainly dominated by *Firmicutes* (99.99% ±0.01%) (Fig 1A). At the genus level, all garum samples were primarily comprised of *Staphylococcus* (71.24% ±23.02), unknown bacteria (10.36% ±13.15) and *Latilactobacillus* (9.56% ±8.45). The garum from intestines consisted of *Staphylococcus* (71.24% ±23.02), unknown bacteria (10.36% ±13.15) and *Latilactobacillus* (9.56% ±8.45), while the garum made from meat consisted of *Staphylococcus* (88.41% ±13.51) and *Latilactobacillus* (8.85% ±11.14) (Fig 1B).

From the 10 most abundant taxa at the phylum level, the crop microbiota is distinguished from both the stomach and intestinal microbiota by having a significantly higher abundance of *Firmicutes* ($p < 0.001$ in both cases) and a higher amount of *Proteobacteria* ($p < 0.001$ in both cases). Furthermore, the crop has a significantly lower abundance of *Desulfobacterota* ($p < 0.001$) compared to the stomach and of *Actinobacteriota* ($p < 0.001$) compared to the intestines. The stomach microbiota is differentiated from the crop and intestinal microbiota by a significantly higher abundance of *Actinobacteria* ($p < 0.001$ and $p = 0.003$, respectively) and *Desulfobacterota* ($p < 0.001$ and $p = 0.039$, respectively) and *Synergistota* ($p < 0.001$ and $p = 0.01$, respectively). Lastly, the intestinal microbiota was distinguished from both the crop

and stomach microbiota by a significantly higher abundance of *Campylobacterota (p < 0.001* in both cases).

At the genus level, the crop microbiota is distinguished from both the stomach and intestinal microbiota, with a significantly higher abundance of *Streptococcus (p < 0.001* in both cases). Furthermore, the crop has a significantly lower abundance of *Olsenella (p < 0.001* in both cases), *Colidextribacter (p < 0.001* in both cases), and *Actinomyces (p < 0.001)* compared to intestines. The stomach microbiota is distinguished from the crop and intestinal microbiota, with a significantly higher abundance of *Actinomyces (p < 0.001* and p = *0.012*, respectively). Finally, the intestinal microbiota is distinguished from both the microbiota of the crop and the stomach by a significantly higher abundance of *Helicobacter (p < 0.001* in both cases).

The crop core microbiota was defined as a subset of ASVs that had a prevalence in over 50% of the samples possessing a relative abundance exceeding 0.0001% of the sample type [43, 44]. The core microbiota of the crop samples was comprised of 5 ASVs, representing 0.62% of the total ASV diversity in the crop microbiota. Notably, 92.3% of the sequenced reads attributed to these core ASVs belonged to the genus *Streptococcus*, a member of the *Firmicutes*, and were detected in 96.6% of the samples. Among the remaining ASVs, three were affiliated with the phylum *Proteobacteria* (4.3%), with one occurring in 63.3% of the samples, and the other two in 53.3% of the samples. Finally, one ASV belonged to the phylum *Cyanobacteria* (3.2%), also prevalent in 53.3% of the samples.

## Alpha-diversity of the ptarmigan gastrointestinal tract and garum

Four indices of alpha diversity were calculated: the Richness, Faith phylogenetic diversity, Shannon index, and Simpson index (Fig 2).

For all alpha diversity metrics, a Shapiro-Wilks test was performed to test if the samples were normally distributed for all four metrics ($p < 0.001$). As none of the metrics were normally distributed, a pairwise Wilcoxon test with "fdr" was used for multiple comparisons to test which of the sample types were significantly different from each other. For the richness and phylogenetic diversity metrics, all sample types were significantly different except for crop

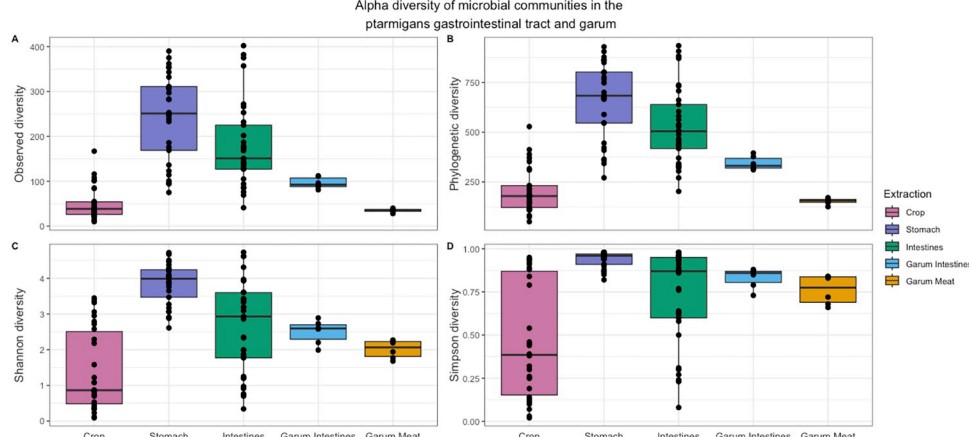

**Fig 2. Comparison of alpha diversity metrics of the microbial communities in the ptarmigan gastrointestinal tract and garum.** Boxplots of the alpha diversity of the ptarmigans' gastrointestinal tract and the two types of garum. The indices used are richness (A), phylogenetic diversity, faith (B), ASV diversity, Shannon index (C), and Simpson index (D). The vertical line inside the boxes is the median obtained from the samples analyzed; each dot represents the alpha diversity value of one sample (Crop, n = 30; Stomach, n = 29; Intestines, n = 33; Garum Intestines, n = 6; Garum meat, n = 6).

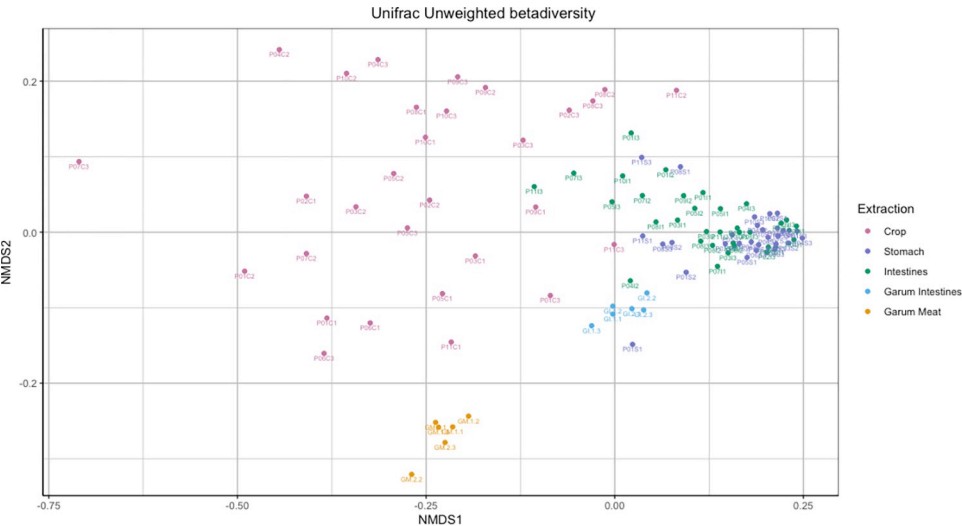

**Fig 3. Non-metric multidimensional scaling of samples from ptarmigan's gastrointestinal tract and garum.**
Bacterial community clustering was assessed based on Unifrac unweighted distance and visualized with a Non-Metric
Multidimensional (NMDS) plot. The NMDS plot displays the distance in the bacterial community composition
between Crop (pink), Stomach (purple). Intestines (green), Garum Intestines (light blue), and Garum Meat (orange).
Each circle represents the ASV composition from one sample (Crop, n = 30; Stomach, n = 29; Intestines, n = 33;
Garum Intestines, n = 6; Garum meat, n = 6).

compared to garum meat. The Shannon and Simpson diversity significantly differed between
the stomach and the crop, intestines, garum intestines, and garum meat ($p < 0.001$, in all
cases) S2 Table.

## Beta diversity of bacterial communities

Beta diversity was assessed using the Unifrac unweighted distance metric in an NMDS plot to
illustrate the differences in the bacterial communities in the ptarmigans' gastrointestinal tract
and the two types of garum (Fig 3). Using PERMANOVA a significant difference was found
between crop and garum intestines ($p = 0.01$, $R^2$ $0.11$), crop and garum meat ($p = 0.01$, $R^2$
$0.14$), crop and intestines ($p = 0.01$, $R^2$ $0.16$), crop and stomach ($p = 0.01$, $R^2$ $0.20$), garum intes-
tines and garum meat ($p = 0.01$, $R^2$ $0.47$), garum intestines and intestines ($p = 0.01$, $R^2$ $0.11$),
garum intestines and stomach ($p = 0.05$, $R^2$ $0.15$), garum meat and intestines ($p = 0.01$, $R^2$
$0.24$) and garum meat and stomach ($p = 0.01$, $R^2$ $0.31$) S3 Table. Furthermore, we see that the
crop had the highest average distance within the samples (0.5515), followed by intestines
(0.45), stomach (0.4235), garum intestines (0.3828), and garum meat (0.2619) (Table 2).

## Integration of intestinal microbiota into garum fermented sauce

Samples from the crop shared 47 and 31 ASVs with the stomach and intestine, respectively (Fig
4A). The stomach and intestines shared 834 ASVs, and 117 ASVs were shared between all three

**Table 2. Betadiversity distance of each sample type.**

| Crop | Stomach | Intestines | Garum Intestines | Garum Meat |
|------|---------|------------|------------------|------------|
| 0.5515 | 0.4235 | 0.4500 | 0.3828 | 0.2619 |

The average distance of the different sample types to their individual centroid indicating how similar each sample type is.

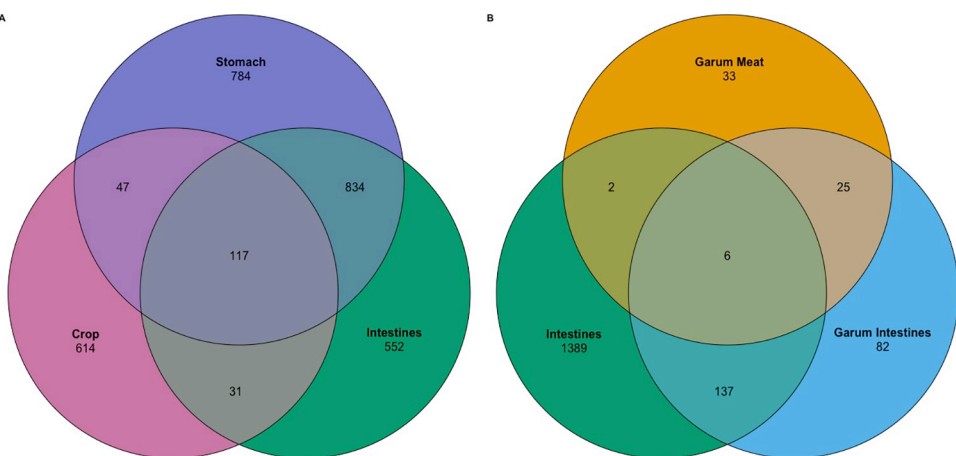

**Fig 4. Shared microbiota between sample types.** Venn diagram shows the number of unique and shared ASVs between sample types. Ptarmigan's gastrointestinal tracts microbiota (Crop, n = 31 (pink); Stomach, n = 29 (purple); Intestines, n = 33 (green)) (A) and between the ptarmigan's intestinal microbiota and the two types of garum (Intestines, n = 33 (green); Garum Intestines, n = 6 (light blue); Garum Meat, n = 6 (orange)) (B).

sample types. When comparing the microbiota of the two different garum types with the intestines, we see that the intestinal microbiota and the garum intestine microbiota shared 137 ASVs, the garum intestines and the garum meat shared 25 ASVs, and the garum meat and intestines only shared 2 ASVs, while 6 ASVs were shared among all three sample types (Fig 4B).

## Taste evaluation

There was no taste difference between Garum Intestines 1 and Garum Intestines 2 nor between Garum Meat 1 and Garum Meat 2 after the initial tasting. The data was combined into one Garum Intestines (1+2 combined) and one Garum Meat (1+2 combined). Results from the CATA were assessed using the Cochran Q test or the Fisher exact test if the Cochran Q test was not applicable. Of the 33 tasting attributes, "Black Olives" and "Game Flavor" were found to have significant differences between the CATA results ($p > 0.001$ and $p = 0.004$, respectively), with garum intestines having significantly higher counts of "yes" in both cases. A complete list of CATA results is listed in S4 Table.

## Discussion

Using a 16S rRNA gene amplicon sequencing strategy, we studied the composition and diversity of the ptarmigans' gastrointestinal microbiota from a culinary perspective. We compared the microbial composition in the ptarmigans' gastrointestinal tract with two types of garum made from the ptarmigans' meat and intestines, respectively. This is the first reported study of the Greenlandic rock ptarmigans' gut microbiota and the first study where the gut microbiota has been put in a food context.

### Microbial composition of the gastrointestinal tract of the rock ptarmigan and its relevance for the human gut microbiota

The microbial composition varies throughout the ptarmigan's gastrointestinal tract. The crop has a richness of 809 ASVs, with a high fraction of the phylum *Firmicutes*, where the genera *Streptococcus* sp. represents the most ASVs, with one highly dominating ASV being the biggest part of the crop's core microbiota. In many avian species, the crop has a pH level of around 5.5

[45], in which most lactic acid bacteria (LAB) classified under the phylum *Firmicutes* have the best growth conditions. Research suggests that the crop microbiota in poultry serves as a natural barrier that limits and kills pathogens [46]. As an example, some strains of *S. cristatus* found in the crop have been associated with reducing the carriage of *Salmonella* in poultry [47]. In addition, LAB is known for its ability to produce lactic acid as a metabolic byproduct, which adds to an acidic environment that inhibits the growth of certain pathogens. Other studies have found a positive relationship between a high abundance of *Firmicutes* in the gastrointestinal tract and the immune system in domesticated chickens [48–50]. In our data, LAB, including *L. delbrueckii* and S. *cristatus*, were identified in the crop microbiota, supporting the idea that the crop of the Greenland rock ptarmigan could have some protective effect against pathogens.

The crop samples exhibited a significantly reduced number of ASVs compared to the stomach and intestinal samples (Fig 2) while having a higher microbial beta diversity (Table 2). Higher beta diversity among the crop samples indicates a less stable microbial community in the crop compared to downstream sites in the intestines. This, alongside that the crop's core microbiota comprised only 5 ASVs, further suggests that the crop's microbiota has a higher fraction of allochthonous microbes. In Fig 4A, we see that the crop samples have a total of 809 ASVs, the stomach samples have 1782 ASVs, and the intestinal samples have 1534 ASVs. Furthermore, we also see in Fig 4A that the microbial composition of the ptarmigan crop shares far less ASVs with either the stomach or the intestines. Taken together, these results might suggest that the microbial community found in the crop could act as a microbial barrier between the outside environment and the lower digestive tract.

Our results from this study differ from previous studies looking into the microbial composition of the ptarmigans' gastrointestinal tract. The dominant phyla found in wild rock ptarmigans from Norway [19] and Japan [20] were *Firmicutes*, *Actinobacteriota*, *Bacteroidota*, and *Synergistetes*. In our data, the dominant phyla were *Campylobacterota* in the intestinal microbiota, followed by *Firmicutes* and *Actinobacteriota*. The genus *Helicobacter* was the most abundant genera after unknown bacteria found in the ptarmigan intestinal microbiota, with 25 different ASVs (Fig 1B). The presence of *Helicobacter* spp. in the intestines could be a source of microbial diversity through gastrophagy from ptarmigan with potential relevance to human health. *H. pylori* has been associated as a cause of diseases and with protective abilities on human health [51, 52]. The bacteria is found in most human gut microbiota, especially in non-industrialized countries, and when present, it is often the dominant microorganism in the stomach [51, 53, 54]. Some *Helicobacter* species are specifically adapted to certain hosts, which may lower the risk of disease transfer to humans [55, 56]. In addition, *H. pylori* is particularly adapted to colonize the human stomach and is thus rarely found in animals, being naturally present only in a limited range of animal hosts [57].

Many factors, including geography, dietary habits, physical activity, and medication usage, influence the microbial composition in the human gut [58]. Dietary transitions in Greenland have meant a divergence from Indigenous animal-sourced Inuit food culture towards more agricultural and processed imported diets [59]. Dietary transitions continuously impact Indigenous peoples' health, and we need a greater understanding of the potential consequences on the human gut microbiota [60]. In the case of the ptarmigan, its integration into non-Indigenous foodways, eating the cooked meat rather than the raw intestinal content, is likely to have resulted in a reduced potential for food microbial diversity. Reduced microbial diversity in the human gut microbiota and its associated negative consequences are more common in Western populations than in populations living more traditional lifestyles [61]. Previous studies show that microbial composition is higher in traditional food in Greenland than in industrial counterparts [15]. Marco and colleagues (2020) suggest that including safe live microbes in our diet

could confer health benefits and should be further investigated. It has been found that diets with live microbes were associated with several health benefits [3, 4]. Consumption of safe microbes has been associated with improved immune systems, reduced risk of chronic diseases, and improved gut function [62, 63]. While the notion of adding probiotics is advocated, it is notable that many probiotic sources contain a relatively limited number of microbial species, and recent studies have argued that not only bacteria labeled as probiotics should be considered beneficial for human health [64, 65]. A recent study done by Hill and colleagues (2023) argues that live microbes from food, including fermented foods, give access to a higher microbial diversity, which could stop the decline of the microbial diversity we see in Western societies caused by processed foods [66, 67] as more and more data point towards that these western diets often have a meager amount of live microbes [68]. Out of the top five phyla identified in this study, four of them (*Firmicutes*, *Actinobacteriota*, *Proteobacteria*, and *Bacteroidota*) are also among the top phyla found in the human gut and may be potential sources of gut microbial diversity.

## Garum, integrating gastrophagy into gastronomy

When comparing the alpha diversity metrics between the two types of garum, garum intestines (GI) and garum meat (GM) (Fig 2), we found that the GI displayed a significantly higher microbial diversity than the GM in both richness and phylogenetic diversity metrics ($p < 0.01$). This suggests that by including intestines and their content in the garum instead of only using meat, we see a higher microbial diversity added to the garum. In other words, by using gastrophagy in our gastronomy, we gain access to a more diverse food microbiota; as we see in Fig 4B, the GI shares 143 ASVs with the ptarmigan intestines while GM shares only 8 ASVs with the ptarmigan intestines. Furthermore, the GI shares more ASVs with the intestines compared to how many ASVs that are not shared. In addition, flavor assessments show that the two garums have distinct flavors. Out of the 33 tasting attributes, "Black Olives" and "Game flavor" exhibited significant differences between GI and GM, with the GI having a higher score for these flavors. The GI has a higher fraction of Other and Unknown genera of Bacteria compared to GM as well as *Actinomyces*, not prevalent in GM. In further studies it would be relevant to test whether the higher diversity of microbes in GI perhaps in combination with the potential of microbes such as *Actinomyces* to metabolize carbohydrates into a variety of acids can help to increase the suppression of pathogens in combination with contributing to allowing access to the richer microbial diversity found in the intestines potentially enhancing the gut health.

Among the shared taxa was the genus *Bifidobacteria*. This family of bacteria, which is naturally found in the human microbiota [69], has been associated with protection from enteropathogenic infection [70] and has been positively associated with treating human diseases [71]. Other taxa identified were the order *Lactobacillales*, where the genera *Lactiplantibacillus* and *Latilactobacillus* were found. In the GI microbiota, 12 and 5 different ASVs belonging to these genera were identified, respectively. For the GM microbiota, only 4 ASVs belonging to this order and 1 ASV for each genus were identified. Both genera are classified as LAB and are associated with a positive impact on human health. For instance, *Lb. plantarum*, which is known to enhance the flavor and preserve the nutritional composition found in food, like amino acids and vitamins. [72]. *Lat. sakei* has been reported to enhance the production of short-chain fatty acids, which are known to improve gut health [73, 74]. Both types of bacteria have been reported to be present in the human gastrointestinal tract and fermented food. Our results show that using gastrophagy—using intestines rather than meat alone—in garum provides access to a significantly broader array of microorganisms with potential relevance to

human health. Introducing gastrophagy into popular culinary practices, such as utilizing garum made from ptarmigan intestines in the kitchen, offers a novel approach to augmenting microbial diversity loss in dishes with added potential flavor implications. This aligns with ongoing efforts to reintegrate diverse microbial exposure into contemporary diets, acknowledging the potential health and sensory benefits associated with such microbes.

Incorporating garum into our project introduces a novel gastronomic dimension to the ptarmigan microbiota. Historically renowned in various cultures, garum exemplifies food preservation through fermentation [75]. This approach acknowledges the adaptability of traditional practices to culinary contexts, emphasizing the resilience and relevance of ptarmigan gastrophagy in present and future foodscapes. Moreover, the exploration of garum as a medium for preserving microbial diversity from ptarmigan gastrophagy expands the scope of our study beyond the biological aspects. It opens avenues for cultural and gastronomic conversations, positioning Indigenous culinary practices as dynamic and adaptable components of a living heritage.

## Conclusions

In conclusion, this study contributes to research at the intersection of traditional diets, human gut microbial diversity, and contemporary gastronomy. It underscores the imperative to acknowledge and preserve the microbial richness inherent in Indigenous dietary practices, advocating for a reassessment of dietary shifts that may compromise this diversity. Ptarmigan gastrophagy is a source of food microbial diversity in a world where the food is becoming less and less microbially diverse. A way to activate the microbial richness found in the ptarmigans' gastrointestinal tract could be through garum or other fermented food practices.

### Future research

Relevant next steps from this research are to assess the impact of eating orunit—ptarmigan intestines—and foods made from it on the human gut microbiome to test whether ptarmigan gastrophagy adds to the diversity of the human gut microbiota and how this impacts human health and wellbeing.

## Supporting information

**S1 Fig. Rarefaction curve of the sample types.** Rarefaction curve of the sample types of crop (A), stomach (B), intestines (C) and the two types of garum meat and intestines (D). The curves confirm a sufficient sequencing depth to cover the microbial diversity found in the samples as they reach asymptote.
(DOCX)

**S1 Table. Tasting attributes for the two garum types used for CATA.**
(DOCX)

**S2 Table. The p-value for the pairwise Wilcoxon test of the alpha diversity indices.** P-value of the alpha diversity metrics richness, faith phylogenetic diversity, Shannon diversity index and Simpsons diversity index. Pairwise Wilcoxon test was performed. P value of 0.05 is set with the p-value adjusted for False Discovery Rate method.
(DOCX)

**S3 Table. PERMANOVA results for the dissimilarity between the bacterial community clustering.** PERMANOVA results for the bacterial community clustering based on Unifrac unweighted distance. P value of 0.05 is set with the p-value adjusted with Bonferroni for

multiple comparisons.
(DOCX)

**S4 Table. CATA results for the two garum types.** P-value of the tasting attributes. The appropriate statistical comparison of 2 groups either Cochran Q test or Fishers exact test was performed. P-value of 0.05 is set.
(DOCX)

## Acknowledgments

The authors would like to extend a special thanks to the hunters who have harvested the ptarmigan. Thanks to Anne Nívika Grødem and Ove Grødem. Furthermore, we would like to thank Ida Broman Nielsen for her work in the laboratory.

## Author Contributions

**Conceptualization:** Diego Prado Vásquez, Aviaja Lyberth Hauptmann.

**Data curation:** Mads Bjørn Bjørnsen.

**Formal analysis:** Mads Bjørn Bjørnsen.

**Funding acquisition:** Aviaja Lyberth Hauptmann.

**Investigation:** Nabila Rodríguez Valerón, Diego Prado Vásquez, Esther Merino Velasco.

**Methodology:** Nabila Rodríguez Valerón, Diego Prado Vásquez, Esther Merino Velasco, Anders Johannes Hansen.

**Project administration:** Aviaja Lyberth Hauptmann.

**Resources:** Nabila Rodríguez Valerón, Diego Prado Vásquez, Esther Merino Velasco, Anders Johannes Hansen.

**Software:** Mads Bjørn Bjørnsen.

**Supervision:** Anders Johannes Hansen, Aviaja Lyberth Hauptmann.

**Visualization:** Mads Bjørn Bjørnsen.

**Writing – original draft:** Mads Bjørn Bjørnsen, Aviaja Lyberth Hauptmann.

**Writing – review & editing:** Mads Bjørn Bjørnsen, Diego Prado Vásquez, Anders Johannes Hansen, Aviaja Lyberth Hauptmann.

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
