## [Decision Letter · Decision Letter 0]

9 Jul 2024

PONE-D-24-21046Microbiota in the ptarmigan intestine - an Inuit delicacy and its potential in popular cuisinePLOS ONE

Dear Dr. Bjørnsen,

Thank you for submitting your manuscript to PLOS ONE. After careful consideration, we feel that it has merit but does not fully meet PLOS ONE’s publication criteria as it currently stands. Therefore, we invite you to submit a revised version of the manuscript that addresses the points raised during the review process.

Both reviewers have provided "accept" recommendation, but I want to make sure that you see their minor comments. A second round of review should not be needed if you address those minor comments.

We look forward to receiving your revised manuscript.

Kind regards,

Franck Carbonero, PhD

Academic Editor

PLOS ONE

Reviewers' comments:

Reviewer's Responses to Questions

**Comments to the Author**

1. Is the manuscript technically sound, and do the data support the conclusions?

Reviewer #1: Yes

Reviewer #2: Yes

2. Has the statistical analysis been performed appropriately and rigorously? 

Reviewer #1: Yes

Reviewer #2: Yes

3. Have the authors made all data underlying the findings in their manuscript fully available?

Reviewer #1: Yes

Reviewer #2: Yes

4. Is the manuscript presented in an intelligible fashion and written in standard English?

Reviewer #1: Yes

Reviewer #2: Yes

5. Review Comments to the Author

Reviewer #1: Overall, the manuscript presents novel information on how dietary microbiota could be a source for gut microbial diversity, and the microbial diversity associated with a traditional food source before and after fermentation. Further, the manuscript explores the gut biogeography of the ptarmigan microbial diversity, which provides additional depth and ecological information on the gut microbiome, something which is lacking in most studies. This manuscript provides several advances to the existing literature on food and gut microbiology.

The Introduction provides helpful context on the sparse published research available in this area, the potential for increased dietary microbial diversity benefits to health, and the food source being evaluated.

The Methods section is detailed and provides information on the sample collection, food preparation, taste evaluation, DNA extraction and sequencing including negative controls, and the QC for the 16S data (which is greatly appreciated). The methods and statistics are appropriate for this approach.

The Results are succinctly presented and easy to follow. The Discussion provides more context to traditional diets and fermented foods, as well the potential implications for health based on the taxa found in this study.

It's very well written, and I only had a few comments regarding formatting.

line 44: there is an open parenthesis but the close is missing

line 261: correct to 53.3% to make it consistent with other percentages

line 275: I'm not sure if Phylogenetic needs to be capitalized

lines 363-366: Helicobacter was found in the intestines, although the species could not be identified. It may be useful here to mention that some species of Helicobactor are adapted to specific hosts, which might reduce the risk of disease in humans. These references may provide useful if the authors decide to include this context: https://www.ncbi.nlm.nih.gov/books/NBK2450/ and https://www.gastrojournal.org/article/S0016-5085(99)70232-5/fulltext

Reviewer #2: The authors investigate the GI microbiome of the Greenland rock ptarmigan whilst comparing it to a garum (traditional fermented sauce) made from both the meat and the intestines to compare the difference in microbial diversity using 16S rRNA sequencing. I found this to be a scientifically sound study that is to the point, well-written and contributes to an important body of knowledge both in the understanding of Indigenous food practice, as well as how gastrophagy can enhance gastronomy through fermentation. Minor comments include:

It would be helpful to provide better labels for Figure 2 so the reader can better follow the alpha diversity analysis (include what alpha diversity measurement was used on the Y-axis and label figures A, B, C, D

Lines 283-285 I prefer to see the R2 in addition to the p-value so the reader can understand the variance explained in the PERMANOVA right away (rather than having to search for it in the table).

Just a thought/suggestion: in lines 395-403 I wonder if a line or two on how/why the fermentation process makes garum an especially safe food for consumption with an increase in microbial diversity utilizing intestines would enhance the argument?

Lines 404-409: another suggestion (but not necessary): you discuss shared taxa between the ptarmigan intestines and the garum intestine. How much of these taxa was shared? It would enhance the paper if you could do a visualization demonstrating how much of bifidobacteria (for example) was detected in the actual intestines and then garum intestines (not for significance of course, but just showing how much of each taxa was detected in the intestines vs garum intestines). As a fermentation person, I would like to know more about this, particularly since these taxa were not shared with the garum made from meat.

6. PLOS authors have the option to publish the peer review history of their article (what does this mean?). If published, this will include your full peer review and any attached files.

Reviewer #1: No

Reviewer #2: No

---

## [Author Response · Author response to Decision Letter 0]

11 Sep 2024

Response to reviewers and editors of PLOS ONE

We thank the reviewers and editors for their time, comments, and constructive suggestions. We provided responses to the reviewers’ comments below.

General revisions:

The formatting of the manuscript has been checked to meet PLOS ONE's formatting requirements. 

The reference list was checked using scite.ai and showed no references with editorial 

concern. 

The data is currently being deposited at the NCBI SRA under ID SUB14679393

Responses to reviewer 1 comments: 

Line 44: there is an open parenthesis but the close is missing – A closing parenthesis has been added at line 45.

Line 261: correct to 53.3% to make it consistent with other percentages – A punctuation (.) has replaced a comma (,) to make it consistent with the rest of the percentages.

Line 275: I'm not sure if Phylogenetic needs to be capitalized – Phylogenetic has been corrected to phylogenetic.

We have included the host-specificity of Helicobacter as recommended by the reviewer and thank the reviewer for relevant discussion point and references. 

The following three references have been added to the reference list in connection to the above revision: 

Dubois A, Berg DE, Incecik ET, Fiala N, Heman-Ackah LM, Del Valle J, et al. Host specificity of Helicobacter pylori strains and host responses in experimentally challenged nonhuman primates. Gastroenterology. 1999;116: 90–96. doi:10.1016/S0016-5085(99)70232-5

Ferrero RL, Jenks PJ. In Vivo Adaptation to the Host. In: Mobley HL, Mendz GL, Hazell SL, editors. Helicobacter pylori: Physiology and Genetics. Washington DC: ASM Press; 2001. pp. 583–592. doi:10.1128/9781555818005.ch46

Lee A, Fox J, Hazell S. Pathogenicity of Helicobacter pylori: a Perspective. Infect Immun. 1993; 1601–1610.

Responses to reviewer 2 comments: 

Figure 2 has been given labels A, B, C, and D and the specific alpha diversity measures have been added at each Y-axis.

R2 has been added to the p-value as suggested.

In response to remark for lines 395-403 a brief section has been added, discussing the potential of the garum to enhance safety of consumption in combination with microbial richness.

We agree that it would be valuable to know more about the shared taxa between the intestines and the garum made with the intestines. We would like to explore this further in a follow-up study that allows for a broader statistical assessment and strain detection.

---

## [Editor Report · Decision Letter 1]

13 Sep 2024

Microbiota in the ptarmigan intestine - an Inuit delicacy and its potential in popular cuisine

PONE-D-24-21046R1

Dear Dr. Hauptmann,

We’re pleased to inform you that your manuscript has been judged scientifically suitable for publication and will be formally accepted for publication once it meets all outstanding technical requirements.

Kind regards,

Franck Carbonero, PhD

Academic Editor

PLOS ONE
---

## [Editor Report · Acceptance letter]

11 Dec 2024

PONE-D-24-21046R1 

PLOS ONE

Dear Dr. Hauptmann, 

I'm pleased to inform you that your manuscript has been deemed suitable for publication in PLOS ONE. Congratulations! Your manuscript is now being handed over to our production team.

Kind regards, 

on behalf of

Dr. Franck Carbonero 

Academic Editor

PLOS ONE